# Engineering of an enhanced synthetic Notch receptor by reducing ligand-independent activation

Zi-jie Yang [1,2], Zi-yan Yu[1,5], Yi-ming Cai[1,6], Rong-rong Du[1] & Liang Cai [1,3,4 ✉]

Notch signaling is highly conserved in most animals and plays critical roles during neurogenesis as well as embryonic development. Synthetic Notch-based systems, modeled from Notch receptors, have been developed to sense and respond to a specific extracellular signal. Recent advancement of synNotch has shown promise for future use in cellular engineering to treat cancers. However, synNotch from Morsut et al. (2016) has a high level of ligand-independent activation, which limits its application. Here we show that adding an intracellular hydrophobic sequence (QHGQLWF, named as RAM7) present in native Notch, significantly reduced ligand-independent activation. Our enhanced synthetic Notch receptor (esNotch) demonstrates up to a 14.6-fold reduction in ligand-independent activation, without affecting its antigen-induced activation efficiency. Our work improves a previously reported transmembrane receptor and provides a powerful tool to develop better transmembrane signaling transduction modules for further advancement of eukaryotic synthetic biology.

[1] Department of Biochemistry, School of Life Sciences and Zhongshan Hospital, Fudan University, 200438 Shanghai, China. [2] School of Basic Medical Sciences, Fudan University, Shanghai, China. [3] State Key Laboratory of Genetic Engineering, Fudan University, Shanghai, China. [4] The Center for Faculty Development of Fudan University, Shanghai, China. [5] Present address: Department of Biology, School of Arts and Sciences, University of Pennsylvania, Philadelphia, PA 19104, USA. [6] Present address: Graduate School of Biomedical Sciences, The University of Texas Health Science Center at Houston, Houston, TX 77225, USA. ✉email: cail@fudan.edu.cn

Customizing cells to sense an extracellular signal and respond accordingly is an important objective in synthetic biology. To this end, a variety of tools capable of transmembrane signaling have been developed, such as the chimeric antigen receptor[1], synthetic Notch[2–4], MESA[5], Tango[6], ChaCha[7], and GEMS[8]. Among these, the synthetic Notch-based system has excellent potential for cancer therapy[9,10] and advancement of synthetic biology[11] due to its high programmability.

First reported in 2016[4], synNotch is composed of an extracellular antigen recognition domain (usually a single-chain variable fragment, scFv), a Notch core regulatory region, and an intracellular domain (ICD). After scFv recognizes the antigen on the sender cells, a conformational change of the negative regulatory region (NRR) in Notch core relays the signal to the Notch transmembrane domain (TMD) also in Notch core (that is NRR + TMD). Sequential conformational changes of the NRR and TMD expose the cleavage sites S2 and S3 to ADAM (a disintegrin, metalloproteinase) and γ-secretase. Proteolytic cleavage releases the ICD, which usually is a transcriptional factor (TF), allowing the triggering of downstream signaling (Fig. 1a).

As reported in several studies[4,9–11], with synNotch, it is necessary to select against cells that display ligand-independent activation (LIA), i.e. that express synNotch against sender cells that do not express the antigen. This clonal selection process is labor-intensive and limits the application of synNotch. Unfortunately, the cause of LIA has yet to be elucidated; understanding of this mechanism is necessary for future applications of synNotch.

Here, we developed a transient co-transfection and flow cytometry analysis procedure to reproduce and study LIA. We found that the high expression of the synNotch receptor correlates positively with LIA. We further showed that adding an intracellular hydrophobic sequence (QHGQLWF) after Notch core significantly reduces LIA of synNotch, without affecting the efficiency of antigen-induced activation efficiency. We confirmed this improvement with multiple variants of synNotch, and named our improved version the enhanced synthetic Notch receptor, esNotch.

## Results

**Ligand-independent activation of synNotch.** We transiently transfected cells with a high amount of synNotch plasmid DNA (Fig. 1b–f) and reproduced the ligand-independent activation (LIA). Cells expressing synNotch were co-transfected with plasmid DNA expressing mCherry, causing them to display red fluorescence detectable by flow cytometry (Fig. 1c, Supplementary Fig. 1). Despite variance in the amount of transfected synNotch plasmid DNA, we were able to consistently obtain 40–60% cells in the population expressing synNotch (Fig. 1c). Using an antibody against the Myc tag present in the extracellular domain of synNotch, we showed that membrane expression of synNotch positively correlated with the amount of DNA transfected (Fig. 1g). Green fluorescence was used as an indicator of LIA. As outlined in Fig. 1d, LIA results in the release of tTAA, which translocates into the nucleus and triggers the expression of a short-lived version of EGFP (d2EGFP). In the absence of antigen-expressing sender cells, this green fluorescence is a direct measurement of LIA. Populations expressing a greater amount of synNotch not only have a high percentage of green cells (Fig. 1e), but also show bright green fluorescence (Fig. 1f). We confirmed this observation in 293T cells stably expressing the same synNotch (Fig. 1h): cells with an increased amount of membrane-expressed synNotch have high LIA.

To demonstrate our synNotch cells could respond normally to their antigen (Fig. 1i), we incubated these cells with their sender cells (Supplementary Figs. 2 and 3) for 24 h. As shown in Fig. 1j, only cells with medium or low synNotch expression responded to their antigen. Due to LIA, cells with high synNotch expression produced comparable levels of green fluorescence, with or without their antigen. A previous study has suggested that LIA can be reduced by extracellularly addition of an EGF repeat (Fig. 1k) to the N-terminus of the Notch core[4]. However, we were not able to reproduce this in our setup (Fig. 1l).

Notch activation relies on the sequential cleavage of S1, S2, and S3[12,13]. To investigate the cause of LIA, we generated S1, S2, and S3 cleavage site mutants[14–16]. We found that while synNotch with either the S1 or S2 mutation had high levels of LIA, synNotch with the S3 mutation had significantly reduced LIA (Fig. 2a). We confirmed this observation by treating cells expressing wild-type synNotch with specific protease inhibitors (Fig. 2b). The use of BB-94, an ADAM inhibitor, did not make a statistically significant difference in LIA. In contrast, the use of compound E, a specific inhibitor of γ-secretase, effectively reduced LIA in a dose-dependent manner. In summary, we reproduced LIA in our setup, and found that antigen-independent S3 cleavage of excessive membrane synNotch could be the cause of LIA.

**Adding a hydrophobic sequence to synNotch suppresses LIA.** A recent structural study of γ-secretase cleavage[17], reported that when native Notch is recognized by γ-secretase, the C-terminus of the TMD must be precisely positioned to form a hybrid β-sheet with the protease. Another structural study suggested that the RAM sequence behind the C-terminus of TMD could form a membrane docking domain[18], which may help the positioning of TMD. Because the RAM sequence was not included in any published synNotch, we hypothesized that adding this natural motif might suppress LIA.

We constructed a synNotch with the RAM sequence (hN1RAM8, $Q^{1763}$HGQLWFP$^{1770}$ from human Notch 1) inserted in the C-terminus of the Notch core (NRR + TMD). We found that this insertion significantly reduced LIA (Fig. 2c). Antibody staining showed that neither the localization nor the expression level of synNotch changed upon RAM insertion (Fig. 2d). We additionally determined the minimal RAM sequence required to suppress LIA (Fig. 2e) as QHGQLWF (hN1RAM7).

To study the effects of individual residues in RAM on LIA, we aligned mouse and human paralogues. The RAM sequence contains a highly conserved hydrophobic tetrapeptide ΦWΦP (Φ = hydrophobic residue) (Fig. 2f). Both the typtophan and phenylalanine are anchored in the membrane[18]. A previous functional study suggested the ΦWΦP tetrapeptide plays an important role in the assembly of the Notch ICD, alongside CSL (an acronym for three homologous proteins, CBF-1/RBPJ-κ in *Homo sapiens*/*Mus musculus* respectively, Suppressor of Hairless in *Drosophila melanogaster*, and Lag-1 in *Caenorhabditis elegans*)[19]. We tested the RAM7 sequences from different Notch paralogues and determined those that significantly reduced LIA (Fig. 2g).

Next, we replaced WΦ at the end of the RAM7 sequence with other hydrophobic residues (FW, FF, WW, and LL, to be specific), and found that synNotch LIA remained suppressed (Fig. 2h). Conversely, replacing WΦ with hydrophilic residues did not suppress LIA. These findings indicated that hydrophobic residues at the end of the RAM7 sequence (Fig. 3a) are required to reduce LIA.

Interestingly, the RAM sequence alone cannot suppress LIA, because synNotch with RAM but not NRR displayed a high level of LIA (Fig. 2i). Thus, NRR and RAM form a NAND logic module for LIA, independent of antigen-induced activation.

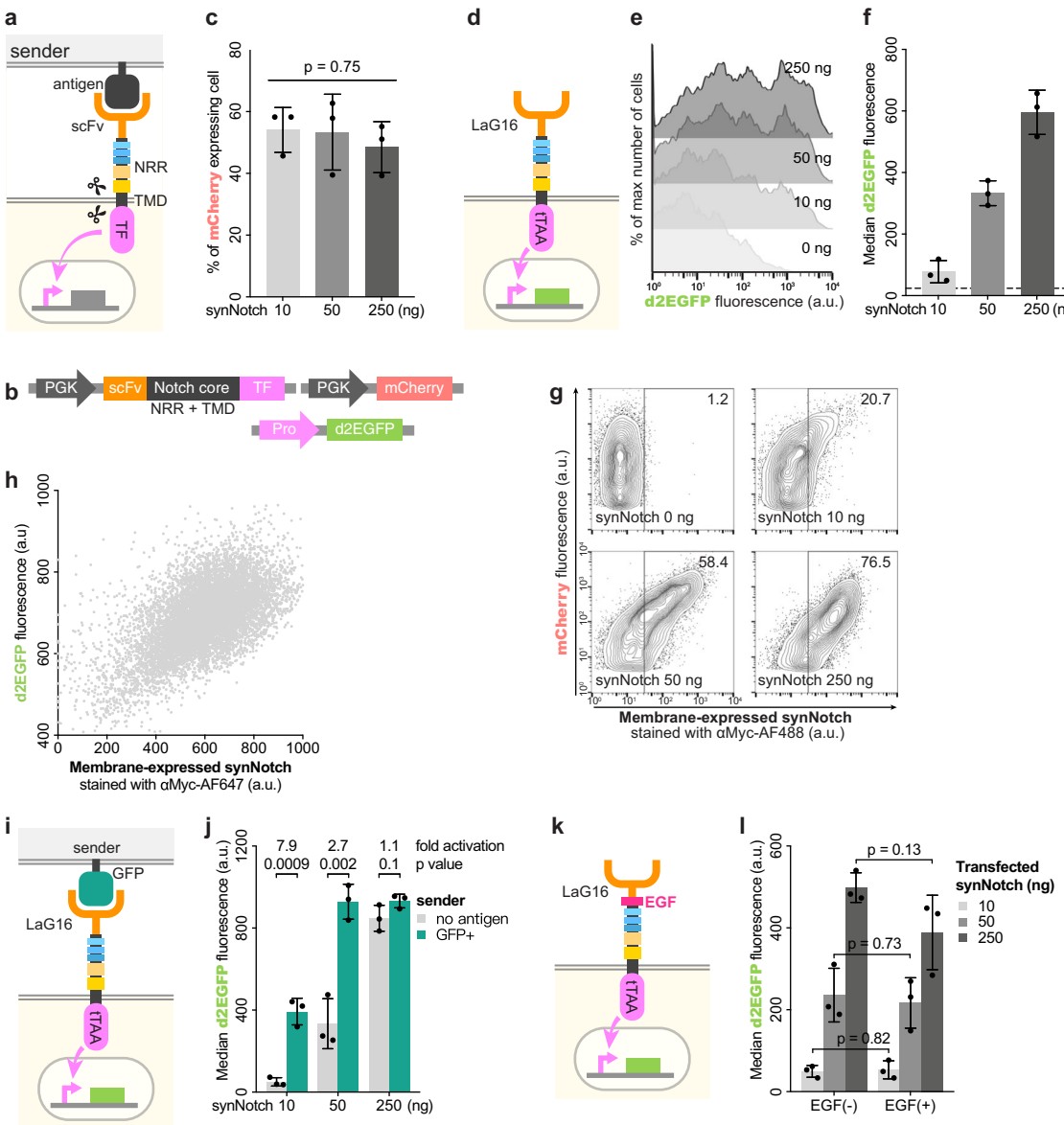

**Fig. 1 Overexpression of synNotch and its relationship to ligand-independent activation (LIA). a** Diagram of antigen-induced synNotch signaling. **b** Cells expressing PGK driven synNotch always express PGK driven mCherry. The matching promoter (Pro) drives the expression of a short-lived version of EGFP (d2EGFP). **c** Co-transfection efficiency measured by mCherry fluorescence. A total amount of 450 ng PGK driven plasmids includes a constant amount of 200 ng PGK driven mCherry and various amounts of PGK driven synNotch. One-way ANOVA for p values. Details on flow cytometry gating in Supplementary Fig. 1. **d** Diagram of synNotch LIA. LaG16 is the nanobody against GFP. tTAA is the transcription factor that induces d2EGFP expression downstream of TRE3G. **e** For experiments as in **d**, stacked histograms show the expression of d2EGFP when different amounts of PGK driven synNotch were transfected. For 0 ng synNotch, 250 ng plasmids that only contain the PGK promoter were used. **f** For experiments in **e**, the median d2EGFP fluorescence intensity of each co-transfection was calculated and presented as a scatter dot plot (bar:mean ± SD). Dashed line, 0 ng synNotch. **g** Cells were co-transfected similar to **e**, **f** but without TRE3G driven d2EGFP, fixed and stained (without permeabilization) with antibody against Myc tag. We inserted a Myc tag between the signal peptide and scFv to facilitate the detection of synNotch expression on the cell membrane. The fluorescence signal for mCherry and extracelluar Myc tag in different conditions are shown. The numbers on the upper right corners of each contour plot are the percentages of Myc-tag positive cells. **h** Cells stably expressing synNotch as in **d** were fixed and stained similar to **g**. **i** The sender cells with the corresponding antigen were incubated with cells expressing the synNotch as in **d**. **j** For experiments as in **i**, fold activation was calculated based on the mean values from synNotch cells co-cultured with sender cells with or without antigen. Two-tailed *t*-test for *p* values. **k** Similar to **d**, cells were co-transfected with synNotch with an EGF repeat inserted between LaG16 and NRR. **l** For experiments in **k**, EGF(−) stands for the synNotch in **d**, while EGF(+) stands for the synNotch in **k**. Two-tailed *t*-test for *p* values.

We subsequently inserted the RAM sequence into different versions of synNotch with varied extracellular or intracellular domains. We showed that the RAM insertion generally suppressed LIA (Fig. 3d, g, j, m), without affecting the efficiency of antigen-induced activation (Fig. 3c, f, i, l, Supplementary Fig. 3). We termed these enhanced synthetic Notch receptors, esNotches.

## Discussion

Specificity is critical for any application of synNotch in basic research or clinical scenarios. Studies using chimeric antigen receptor (CAR) T cell therapy have demonstrated collateral damage due to off-target effects of the engineered receptor[20]. Our results suggest that high expression of synNotch correlates

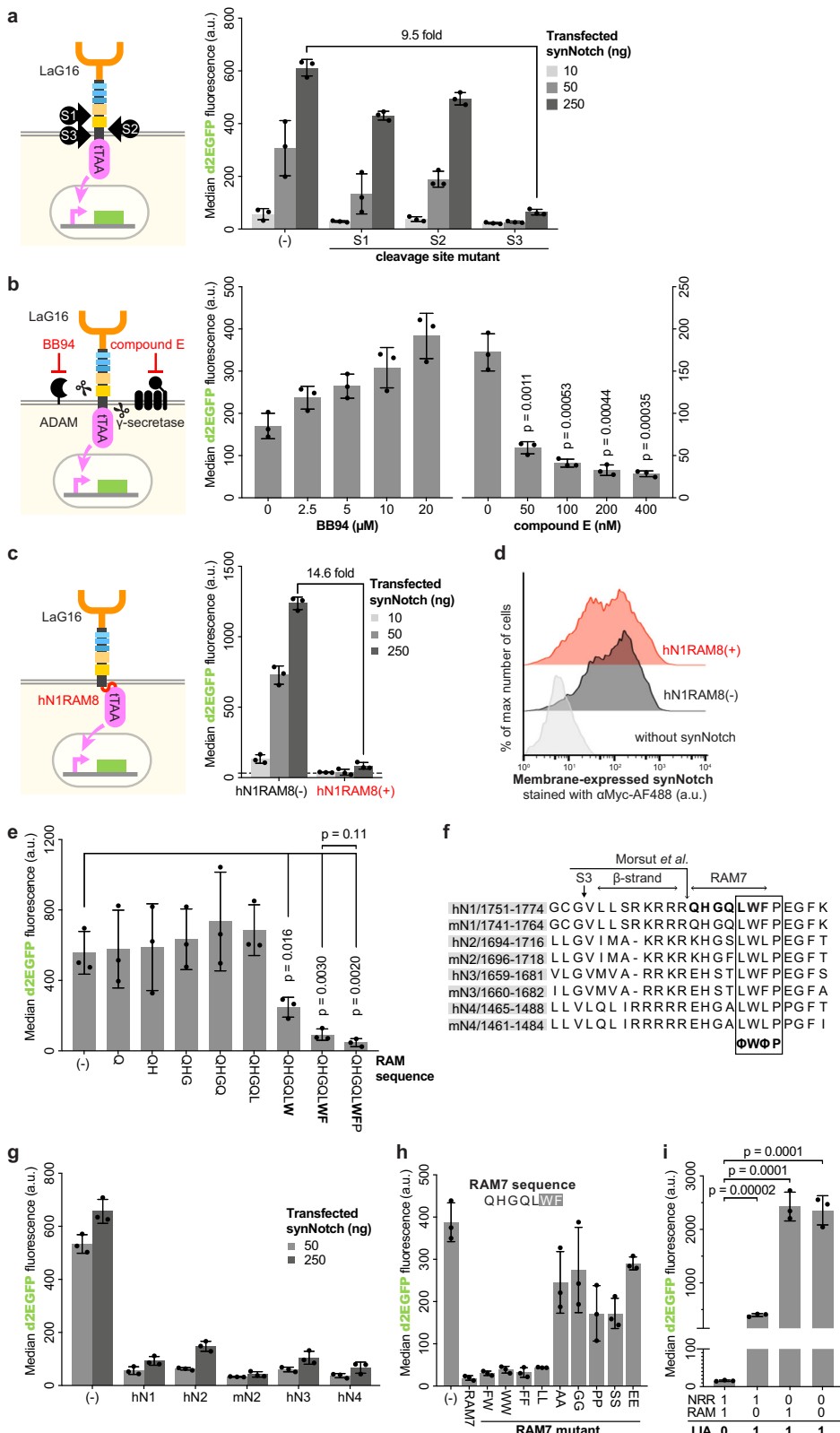

with high ligand-independent activation (LIA). We anticipate that cells carrying synNotch with high LIA could cause clinical complications when used in vivo. The clonal selection process used by Morsut et al., which produce patient-specific synNotch cells, does not provide a practical solution as it is laborious and inefficient. Here, we reported that adding an intracellular hydrophobic sequence to synNotch suppresses

LIA without affecting antigen-induced activation. We named our synthetic Notch receptor incorporating this sequence the enhanced synthetic Notch receptor, esNotch (Fig. 3a). In the context of the around 326 total amino acids that make up the entire Notch core[4], the seven hydrophobic amino acids (RAM7) we inserted should not cause difficulty in manufacturing esNotch cells.

**Fig. 2 Suppression of LIA and contributions of RAM residues to suppress LIA. a** Similar to Fig. 1d, cells were co-transfected with synNotch with mutations at S1, S2 or S3 proteolytic cleavage sites. The median d2EGFP fluorescence intensity of each co-transfection was calculated and presented as a scatter dot plot (bar:mean ± SD). Group (−) is the synNotch as in Fig. 1d. Fold change was calculated based on the mean values from 250 ng co-transfected synNotch cells without or with S3 mutation (GCG V → LLFF). **b** Similar to Fig. 1d, cells were co-transfected with synNotch and incubated with different concentrations of protease inhibitor. Two-tailed t-test for p values. **c** The RAM sequence presented in human Notch 1 (hN1RAM8 for short) was inserted into the synNotch as in Fig. 1d, between TMD and tTAA. Fold change was calculated based on the mean values from 250 ng co-transfected synNotch cells without or with hN1RAM8. **d** Cells were co-transfected as in **c** but without TRE3G driven d2EGFP, fixed and stained (without permeabilization) with antibody against Myc tag in synNotch. Stacked histograms show the fluorescence signal for Myc tag under different conditions. **e** Similar to **c**, different RAM sequences were used for the experiments. Two-tailed t-test for p values. **f** Alignment of the amino acid sequences near mouse and human Notch receptors' TMD. S3 marked the γ-secretase cleavage site. The arrow with labeling Morsut et al. marks the end of TMD. β-strand is required to form the critical β-sheet reported in a recent structural study[17]. **g** Similar to experiments in Fig. 1d, different RAM7 sequences as in **f** were inserted into synNotch between TMD and tTAA. Group (−) is the synNotch as in Fig. 1d. **h** Similar to experiments in Fig. 1d, different amino acid sequences were inserted into synNotch between TMD and tTAA. Group (−) is the synNotch as in Fig. 1d. **i** Similar to experiments in Fig. 1d, synNotch with or without NRR and/or RAM were used for co-transfection. The median d2EGFP fluorescence intensity of each co-transfection was calculated and presented as a scatter dot plot (bar: mean ± SD).

In cells expressing esNotch, there is a notable dampening in the absolute magnitudes of the responses, with or without the antigen. It might be due to adding RAM7 positions the TMD less prone to γ-secretase cleavage. As a solution, esNotch may be limited if its ICD does not act as a transcriptional factor. However, an otherwise undetectable level of the Notch 1 receptor ICD is sufficient to induce downstream signaling[21]. This argues that the Notch receptor is an initiator of transmembrane signaling rather than its executor, which applies for synthetic Notch receptors as well. Once signaling is initiated and the antigen-induced transcripts are produced, the downstream effect is generated after translation, during which the lifetime of transcripts and/or protein products determine the response magnitude.

Because NRR and RAM form a NAND logic module for LIA (Fig. 2i), we also attempted to reduce LIA by mutagenizing the NRR. This approach failed to reduce LIA, whether involving a single amino acid replacement or a short deletion.

A highly specific synthetic Notch receptor has great potential for advancing synthetic biology. Firstly, esNotch could be used to develop better transmembrane signaling transduction modules. Existing synthetic receptors are able to transduce extracellular signals into the membrane, but their downstream activation usually appears as ALL or NONE. In contrast, native transmembrane proteins show a great variety of downstream activation patterns. For example, the size of the ligand-receptor contact area affects Notch signaling and directs cell fate decisions[22]. Using an engineered transcriptional system, we have demonstrated the ability of synNotch to respond to the antigen at a range of densities (Fig. 3n, more details at http://2017.igem.org/Team:Fudan), which brings us one step closer to the goal of context-specific responses.

Secondly, multiple esNotch constructs could be used together to create binary transmembrane logic gates (Fig. 3o, more details at http://2018.igem.org/Team:Fudan), the building blocks of a complex signal transduction network. Due to LIA, our initial attempts to build XOR and XNOR logic gates with synNotch failed. Hopefully, such combinatorial interactions[23] will be feasible with esNotch.

Thirdly, esNotch could be used together with other engineered receptors, such as CAR[9,24], to generate temporally separated responses. Toda et al. demonstrated the promising progress on this front by manipulating the sequential activation of synNotch and regulating the resulting self-organizing patterns[11].

Synthetic biology applies engineering principles to build artificial biological tools for research, manufactory and medical applications. Our work in this study improved a previously reported transmembrane receptor by reducing ligand-independent activation, thereby improving the receptor's performance. Our improved receptor, esNotch (Fig. 3a), is a powerful tool for the further advancement of synthetic biology.

## Methods

**Molecular cloning.** All constructs were verified by Sanger sequencing. Briefly, synNotch receptors were built by fusing the scFv (LaG16, αCD19, or αHer2), the Notch core regulatory region (mN1c$^{NP\_032740, \text{Ile}1427-\text{Arg}1752}$, EGF-mN1c, hN1c$^{NP\_060087, \text{Ile}1427-\text{Arg}1762}$ or Notch core with RAM sequence), and the transcription factor (tTAA or CV2). tTAA would bind to TRE3G with 7 copies of TetO (TCCCT ATCAG TGATA GAGA), while CV2 (CymR-VP64) would bind to pCuO with 6 copies of CuO (AAACA GACAA TCTGG TCTGT TT). All synNotch constructs contained an N-terminal CD8α signal peptide (MALPV TALLL PLALL LHAAR P) for membrane targeting and a Myc tag (EQKLI SEEDL) for determination of surface expression. All synNotch constructs are summarized in Supplementary Table 1. Detailed sequences are available upon request (email should be addressed to L.C.).

**Cell culture.** In all, 293T cells were cultured in DMEM supplemented with 10% FBS (HyClone), 100 U/ml penicillin, 100 μg/ml streptomycin and 1x GlutaMax (Gibco). K562 cells were cultured in RPMI 1640 supplemented with 10% FBS, 10 mM HEPES, 100 U/ml penicillin, 100 μg/ml streptomycin and 1x GlutaMax. Cells were cultivated at 37 °C in a humidified atmosphere containing 5% CO$_2$.

**Transient transfection.** Cells were plated onto 24-well plates the day before transfection, such that the cells were 60–80% confluent in each well on the day of transfection. Transient transfections were performed using PEI MAX (Polysciences #24765–1). PEI stock solution was made at a concentration of 1 μg/μL in double-distilled water, adjusted to pH to 7.0 and sterilized through filtration (0.22 μm). PEI stock solution was stored at 4 °C and warmed to room temperature before usage.

For each well, a total of 700 ng mixed plasmid DNA was dissolved in 25 μL Opti-MEM (Gibco) and mixed with 25 μL of PEI in Opti-MEM (23.5 μL Opti-MEM, 1.5 μg PEI). The DNA/PEI/Opti-MEM mixture was incubated at room temperature for 10 min and then dropped onto the cells.

Protease inhibitor BB-94 (Sigma-Aldrich, Cas# 130370–60–4) was dissolved in DMSO at a concentration of 10 mM, and compound E (Millipore, Cas# 209986–17–4) at 200 μM. These inhibitor solutions were aliquoted and stored at −20 °C. For experiments using these inhibitors, the solutions were applied 4 h after DNA/PEI/Opti-MEM mixture was dropped onto the cells.

**Stable cell line generation.** To generate cell lines stably expressing synNotch, we constructed one plasmid with all components as in Fig. 1b, flanked by two PiggyBac transposon-specific DNA sequences. We used the synNotch as in Fig. 1d.

As described previously[25], 293T cells were seeded at a density of $2 \times 10^5$ per well in a 12-well plate and transfected the following day with 0.5 μg PiggyBac-based synNotch plasmid and 0.5 μg PiggyBac transposase plasmid (pCMV-PBase). Approximately 10 days after transfection, cells with stable mCherry expression were sorted using a BD FACSJazz.

**Flow cytometry.** Cells were analyzed 48 h after transfection (unless otherwise specified) by flow cytometry on a BD FacsJazz. Briefly, fluorescent cells were dissociated using 0.05% Trypsin-EDTA, fixed using 4% PFA in PBS, washed by PBS, and stored at 4 °C in the dark prior to analysis. To detect surface expressed synNotch or antigen, cells were resuspend using PBS, fixed and stained with αMyc AF488 (Cell Signaling Technology #2279, diluted 1:100), αMyc PE (Cell Signaling Technology #3739, diluted 1:100) or αMyc AF647 (Cell Signaling Technology #2233, diluted 1:100).

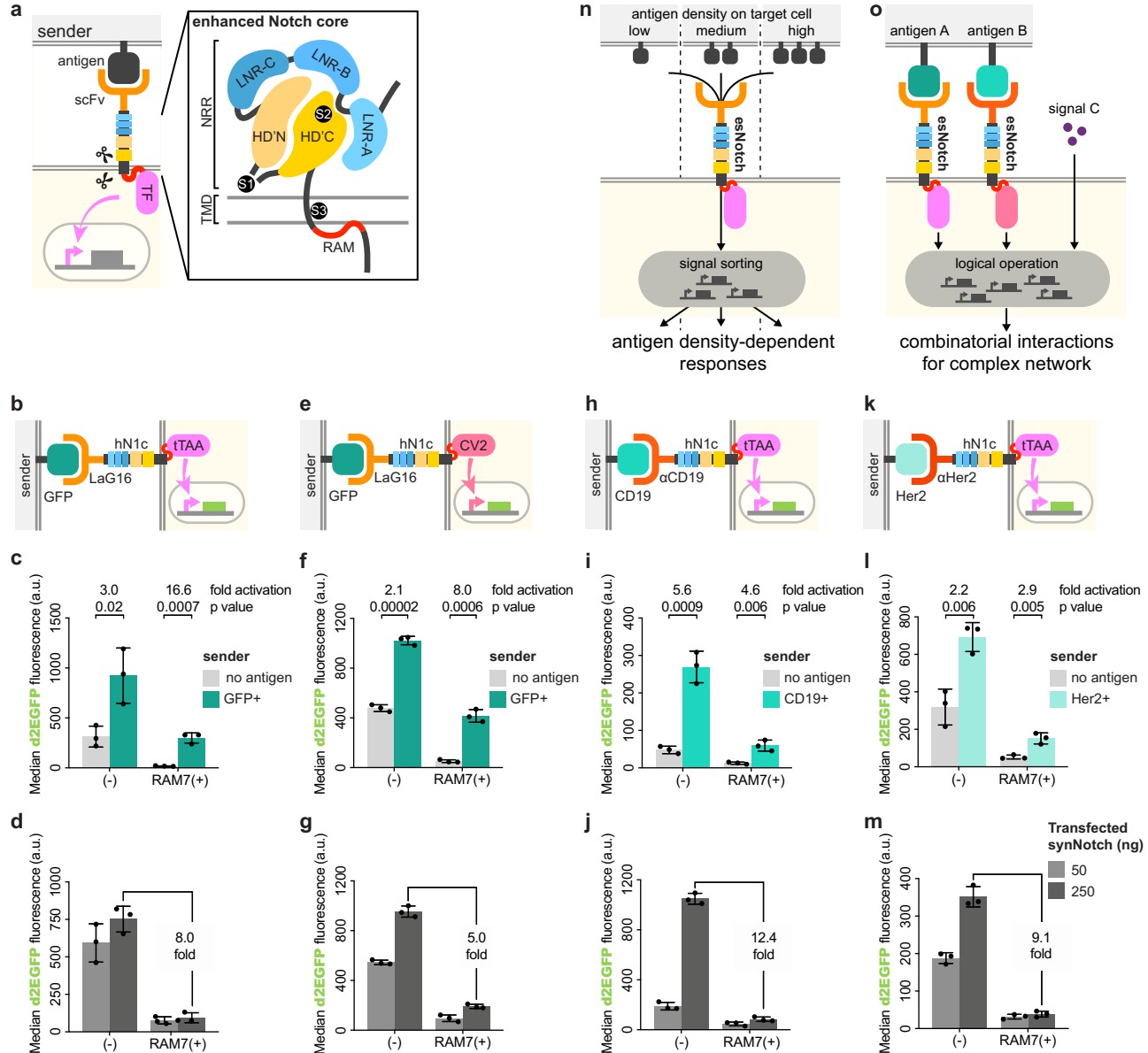

**Fig. 3 Validation and potential applications of an enhanced synthetic Notch receptor. a** Diagram of esNotch with an enhanced Notch core. Abbreviation: scFv, signal-chain variable fragment. TF transcription factor. NRR negative regulatory region. TMD transmembrane domain. LNR-A, LNR-B, LNR-C, HD'N, HD'C are sub-structures of NRR. S1, S2, and S3, the three proteolytic cleavage sites critical for Notch signaling[12,13]. **b** Diagram of a synNotch similar to Fig. 1i, but with hN1RAM7 inserted between TMD and tTAA, as well as replacing mouse Notch core with human Notch core. **e** Diagram of a synNotch similar to **b**, but using CV2 as the transcription factor. Because transcription factor was changed from tTAA to CV2, pCuO driven d2EGFP was used. **h** Diagram of a synNotch similar to **b**, but using αCD19 as the scFv. **k** Diagram of a synNotch similar to **b**, but using αHer2 as the scFv. **c, f, i, l** For experiments as in **b, e, h, k**, the median d2EGFP fluorescence intensity of each co-transfection was calculated and presented as a scatter dot plot (bar: mean ± SD). Fold activation was calculated based on the mean values from synNotch cells co-cultured with sender cells with or without antigen. Groups marked by (−) were co-transfected with synNotch without the RAM sequence. **d, g, j, m** Similar to experiments in **b–e**, but without the sender cells, the median d2EGFP fluorescence intensity of each co-transfection was calculated and presented as a scatter dot plot (bar:mean ± SD). Fold change was calculated based on the mean values from 250 ng co-transfected synNotch cells without or with hN1RAM7. **n** Application of esNotch to generate antigen density-dependent responses. **o** Application of esNotches to construct complex network that could logically respond to multiple extracellular inputs.

Fluorescence from d2EGFP or AF488 was measured by 488 nm excitation laser and 530/40 emission filter, PE was measured by 561 nm laser and 585/29 filter, mCherry was measured by 561 nm laser and 610/20 filter, and AF647 was measured by 633 nm laser and 660/20 filter. We used the forward scatter (FSC) and side scatter (SSC) signals to remove debris (Supplementary Fig. 1). Fluorescence from the different channels were gated and analyzed by FlowJo (version 10, Tree Star), and available at Mendeley (https://doi.org/10.17632/xkfjrgf535.1). At least 20,000 events were measured for each set of data, and at least 50,000 events were analyzed when synNotch cells were co-cultured with sender cells.

**Co-culture synNotch cells with sender cells**. On day 1, synNotch and antigen were transfected into different populations of 293T cells. On day 2, synNotch cells and antigen-expressing sender cells from each well were directly resuspended in ~500 μL fresh media separately, with the same cell counts per μL. Then, 100 μL of synNotch cells were mixed with 400 μL of sender cells and placed in a new well for co-culture. The 24-well plate was centrifuged briefly at 500 g for 3 min before returning to 37 °C. On day 3 (48 h after transfection), the cells were fixed for flow cytometry. We repeated most experiments using K562 cells stably expressing antigens as the sender cells and obtained similar levels of antigen-induced activation (data available at Mendeley https://doi.org/10.17632/xkfjrgf535.1).

**Statistics and reproducibility**. All statistical analyses were performed using FlowJo (version 10, Tree Star), Excel for Mac (version 15.41, Microsoft) and Prism (version 7, GraphPad). All experiments were independently performed in triplicate; unless otherwise indicated. Images were combined and annotated in Illustrator (version CC 2014, Adobe) for presentation.

**Reporting summary**. Further information on research design is available in the Nature Research Reporting Summary linked to this article.

## Data availability
All relevant data were made publicly available at Mendeley (https://doi.org/10.17632/xkfjrgf535.1).

Detailed DNA sequences are available upon request (email should be addressed to L.C.).

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

## Acknowledgements
We acknowledge Stephen C. Blacklow (Harvard) for critical comments on the paper. L.C. was supported by the National Natural Science Foundation of China grants (31222019), and funds from Fudan University, State Key Laboratory of Genetic Engineering and YZT Foundation. The iGEM Team:Fudan was supported by funds from YF Capital and the National Top Talent Undergraduate Training Program.

## Author contributions
Z.-J.Y. and L.C. conceived and designed the experiments. Z.-J.Y., Z.-Y.Y., Y.-M.C., and R.-R.D. performed experiments. Z.-J.Y. and L.C. analyzed the data, wrote and edited the paper.

## Competing interests
L.C. and Z.-J.Y. have the following competing interests: a provisional patent application (ZL2019109554299) about the sequences and applications of esNotch has been filed on 2019 Oct 9th in China. Z.-Y.Y., Y.-M.C. and R.-R.D. declare no competing interests.
