## [Peer Review File · Communications Biology]

Reviewers' comments:

Reviewer #1 (Remarks to the Author):

The authors have shown improvement of Synthetic Notch system by decreasing ligand independent activation via additional protein engineering (adding an intracellular hydrophobic sequence). Enhanced synthetic Notch receptor (esNotch) performs better showing higher ON-OFF functions. This study can add a new (and important) step towards development of effective sensing-response elements in synthetic biology and can potentially improve the SynNotch practice in clinical settings. Please find below my comments:

In highlight section authors mention "Validation and applications of an enhanced synthetic Notch receptor, esNotch", but the "application" is just at a level of discussion and doesn't have any data therefore should be removed from highlight.

Fig 3 legend mentions "replacing mouse Notch core with human Notch core" is this the same hydrophobic sequences, if yes, please clarify and also in methods please address. If this is a different alteration please clarify in the text, explain beneficial effect and compare mouse vs human versions. When using CD19 the on/off ratio is not as good compared with GFP (Fig 3 a',b' versus Fig 3 c'). What's the reason? It is not clear whether the esNotch improvement is exclusive to GFP sensing or can be applicable with other antigens as CD19 sending doesn't show improvement. Thus, it is key for authors show brief data testing a third type of antigen or provide convincing explanation. For instance, is this related to number of senders?

Please show a more comprehensive dynamic data for temporal effect of SynNotch -TF vs esNotch-TF for activation effect on a reporter such as EGFP. How over time newly green cells may be generated (population dynamic of cells) and also intensity per cells will be amplified. Would esNotch shows a comparable amplified signal as compared with SynNotch overtime (as the authors claimed in 2nd paragraph in discussion)?

Please provide how gating is done for FACS analysis and provide this data in supplementary.

Please add statistical analysis where applicable.

When performing experiments what's ratio between sender and receiver can control the performance of the system. Would the authors have data that can help clarify that and also I can't find what ratios were assessed in this study.

In discussion, "Using an engineered transcriptional system, we have demonstrated the ability of synNotch to respond to the antigen at a specific density (Fig. 3d, more details at xxxx), which brings us one step closer to the goal of context-specific responses.

It is not clear how improvement of specificity can deliver "antigen-density dependent response" pointed out in Fig 3d. It should be discussed better or should be removed.

Fig 3f: "Schematic diagram of esNotch with an enhanced Notch core" should be moved to earlier and a comparable image of SynNotch should be included.

Reviewer #2 (Remarks to the Author):

This is an excellent, short and straightforward paper. It shows that the leaky activity of synNotch, a synthetic modular receptor allowing for programmable cell-cell interactions, can be suppressed by including a short hydrophobic sequence present in native Notch while retaining inducibility. This sequence is predicted to insert into the inner leaflet of the plasma membrane and to help position the enzymatic complex that processes synNotch upon ligand-binding. This is an important finding given the huge interest in using synNotch for synthetic biology and applying this molecular device for

therapeutic purposes. The study is technically sound, the results are clear and convincing, the paper is very well written. I strongly support publication.

Dear Reviewers,

Thanks a lot for your constructive comments to our manuscript. In the revised version, we have addressed all of them. Point-by-point response as following:

In highlight section authors mention “Validation and applications of an enhanced synthetic Notch receptor, esNotch”, but the “application” is just at a level of discussion and doesn't have any data therefore should be removed from highlight.

Yes. We have removed it from the highlight. And, we have changed Fig. 3 title into “potential applications”.

Fig 3 legend mentions “replacing mouse Notch core with human Notch core” is this the same hydrophobic sequences

There are 10 amino acids difference: mouse Notch1 core is from NP_032740, between Ile1427 and Arg1752, human Notch1 core is from NP_060087, between Ile1427 and Arg1762. We have included this information in the methods section.

When using CD19 the on/off ratio is not as good compared with GFP. What's the reason?

As suggested by structural studies: (1) the C-terminus of TMD must be precisely positioned for the γ -secretase (Yang *et al* 2019); (2) RAM could form a membrane docking domain (Choi *et al* 2012). We believe adding RAM7 positions the TMD less prone to the protease cleavage (reduced LIA). Besides the dampening of the absolute values, we did see a slight decrease of activation (from 5.6-fold to 4.6-fold) when the antigen is CD19. We think the antibody and its structural change due to antigen binding both affect the positioning of TMD, thus protease cleavage. In the case of CD19 and α CD19 we used, intracellular membrane docking via RAM7 might slightly reduce the efficiency of protease cleavage upon antigen binding.

It is not clear whether the esNotch improvement is exclusive to GFP sensing or can be applicable with other antigens as CD19 sensing doesn't show improvement. Thus, it is key

for authors show brief data testing a third type of antigen or provide convincing explanation.

Please note that esNotch significantly reduces LIA (Fig. 3b''-e''): for GFP, an 8 fold reduction; for CD19, a 12.4 fold reduction; for Her2 (we test a third antigen as suggested), a 9.1 fold. In the case of GFP, we observed not only a decrease of LIA, but also an increase of activation efficiency upon antigen binding (from 3.0-fold to 16.6-fold). In the cases of CD19 and Her2, the activation efficiencies did not significantly change (CD19: 5.6-fold vs. 4.6-fold, Her2: 2.2-fold vs. 2.9-fold).

For instance, is this related to number of senders?

The ratio of senders vs. receivers (esNotch or synNotch cells) is 4:1. Please notice these are 293T cells co-culture experiments following transient transfections, and no additional centrifugation to facilitate cell-cell interactions.

Please show a more comprehensive dynamic data for temporal effect of synNotch -TF vs esNotch-TF for activation effect on a reporter such as EGFP. How over time newly green cells may be generated (population dynamic of cells) and also intensity per cells will be amplified.

In Fig. S2, we show that d2EGFP signal after co-culture of sender cells and synNotch or esNotch cells for 24, 48, and 72 hours. The absolute values peak at 48 hours after co-culture, which is about 72 hours after PEI based transient transfection.

Would esNotch shows a comparable amplified signal as compared with SynNotch overtime (as the authors claimed in 2nd paragraph in discussion) ?

No. We show in Fig. S2 that esNotch does not provide a comparable amplified signal as compared with synNotch after 48 and 72 hours of co-culture. We think the notable dampening in the absolute magnitude is due to the positioning of TMD, which is less prone to protease cleavage after adding RAM7. It is useful for reducing LIA, the major

improvement of esNotch. We have changed the text in the discussion to reflect the new data.

Please provide how gating is done for FACS analysis and provide this data in supplementary.

Yes. In Fig. S1, we have included the gating details for Fig. 1c,1e,1f. Most data in the manuscript are presented in the format of Fig. 1f. We have uploaded all FCS files to a public data depository and will be openly available to the community.

Please add statistical analysis where applicable.

Yes. We have done it.

When performing experiments what's ratio between sender and receiver can control the performance of the system. Would the authors have data that can help clarify that and also I can't find what ratios were assessed in this study.

The ratio of senders vs. receivers (esNotch or synNotch cells) is 4:1. Please notice these are 293T cells co-culture experiments following transient transfections, and no additional centrifugation to facilitate cell-cell interactions. We included this information in the methods section. Both senders and receivers are mixed for 24 hours before sorting. Too many senders in the cell suspension increases the sorting difficulty. We did not test whether 8:1 or 16:1 would increase activation fold after co-culture.

In discussion, "Using an engineered transcriptional system, we have demonstrated the ability of synNotch to respond to the antigen at a specific density (Fig. 3d, more details at xxx), which brings us one step closer to the goal of context-specific responses. It is not clear how improvement of specificity can deliver "antigen-density dependent response" pointed out in Fig 3d . It should be discussed better or should be removed.

The transcriptional network we built previously could generate high/medium/low responses to the same antigen. Per DBPR, we could not reveal the reference of the study. To clear up,

we have changed the text into “antigen at a range of densities”. Based on our previous study, we know that a tighter receptor is better than a stronger receptor. Our esNotch with reduced LIA, a tighter receptor.

Fig 3f: “Schematic diagram of esNotch with an enhanced Notch core” should be moved to earlier and a comparable image of SynNotch should be included.

We now have the schematic diagram in Fig. 3a. The RAM sequence we added is red in the esNotch diagram.

Thank you again for your constructive suggestion.

REVIEWERS' COMMENTS:

Reviewer #1 (Remarks to the Author):

The authors have addressed most of my concerns.

ONE MINOR COMMENT:

for clarity please avoid abbreviations in "highlight" section.